# Learning Management System-Based Evaluation to Determine Academic Efficiency Performance

**Brenda Juárez Santiago** [1,2,†] , **Juan Manuel Olivares Ramírez** [2,*,†],

**Juvenal Rodríguez-Reséndiz** [3,*,†], **Andrés Dector** [4,†], **Raúl García García** [2,†],

**José Eli Eduardo González-Durán** [5,†] and **Fermín Ferriol Sánchez** [1,†]

1.  Facultad de Ingeniería, Universidad Internacional Iberoamericana, Campeche 24560, Mexico; bjuarezs@utsjr.edu.mx (B.J.S.); fermin.ferriol@unini.edu.mx (F.F.S.)
2.  Ingeniería, Universidad Tecnológica de San Juan del Río, Querétaro 76800, Mexico; rgarciag@utsjr.edu.mx
3.  Facultad de Ingeniería, Universidad Autónoma de Querétaro, Querétaro 76010, Mexico
4.  CONACYT–Universidad Tecnológica de San Juan del Río, Querétaro 76800, Mexico; adector@conacyt.mx
5.  Instituto Tecnológico Superior del Sur de Guanajuato, Guanajuato 38980, Mexico; je.gonzalez@itsur.edu.mx
*   Correspondence: jmolivaresr@utsjr.edu.mx (J.M.O.R.); juvenal@uaq.edu.mx (J.R.-R.)
†   These authors contributed equally to this work.

**Abstract:** At present, supporting e-learning with interactive virtual campuses is a future goal in education. Models that measure the levels of acceptance, performance, and academic efficiency have been recently developed. In light of the above, we carried out a study to evaluate a model for which architecture design, configuration, metadata, and statistical coefficients were obtained using four Learning Management Systems (LMSs). That allowed us to determine reliability, accuracy, and correlation, using and integrating the factors that other researchers have previously used, only using isolated models, such as Anxiety–Innovation (AI), Utility and Use (UU), Tools Learning (TL), System Factors (SF), Access Strategies (AS), Virtual Library (VL), and Mobile Use (MU). The research was conducted over one year in nine groups. The results from an LMS Classroom, architecturally and configuration-wise, had the highest level of performance, with an average of 73% when evaluated using statistical coefficients. The LMS Classroom had a good acceptance and a greater impact: SF, 82%, AI, 80%, and VL, 43%, while out of the seven factors, those with the most significant impact on academic efficiency were TL, 80%, VL, 82%, and MU, 85%.

**Keywords:** learning management system; integrating information communication technologies; educational management; e-learning; academic efficiency

## 1. Introduction

Nowadays, Information and Communication Technologies (ICTs) have become a strategy for universities to contribute to overall student performance [1]. The development of these innovative teaching–learning processes have generated in students what it is currently known as Personal Learning Environments (PLEs) [2]. Through the use of ICTs, students now have the opportunity to study off-campus, without the need for an actual teacher–student encounter, using just an e-learning model [3], which also serves as a way to expedite the learning process [4]. ICTs have excellent advantages for academic applications that can be offered through a Learning Management System (LMS), since they facilitate the administration of courses in universities and training centers of organizations, even despite the controversy over how online content is evaluated, as described in [5]. In this context, several studies have been carried out to identify the ability to create a dynamic Virtual Learning Environment (VLE) [6] compatible with the diversities of learners' interactions [7].

These advances have been implemented and tested in what is called the third generation of LMSs, which refers to "[t]he strategy of using technological resources through Web 2.0, which plays an important role for the use of resources through ICT, external to the LMS, and [affords] students greater interaction with the e-learning model" and "[t]hrough the use of technology the generation of new learning environments has been allowed, which facilitate the performance of [academic] activities [...] and improve performance" [8–10], taking into account how the design choices impact the efficiency and legal compliance of personal data protection means [11]. At the University of Chile, a study measured the level of educational and technological use of Moodle and its implications for teaching using a quantitative approach, applying a questionnaire to a sample of 640 teachers of higher education [12].

Several studies were carried out in six different universities in Saudi Arabia to determine the importance of people's attitudes when they use LMSs in the teaching–learning process, showing that attitude is the main barrier to overcome in the implementation of an LMS [13]. In India, there has been some progress; its universities have adopted the LMS system to take full advantage of 700 virtual tools [14].

Humanante conducted an assessment in the computer engineering program in the National University of Chimborazo, Ecuador [2], while Ahmad followed suit and evaluated the learning tools for personal virtual environments and factors influencing the environment in the University in Jordan [15]. At the University of China, the confirmation and perception of the students and the data of 45 Chinese participants were analyzed in an executive postgraduate program, resulting in a conceptual construction model to explain the individual differences between students in the level of acceptance and the use of their VLE [6]. In Iraqi universities, LMSs have been adopted to improve the educational processes in higher education institutions through the Technology Acceptance Model (TAM) [16]. The TAM is a model that determines the factors of the use of ICT, and it is based on the Theory of Reasoned Action (TRA) [17,18] and the Unified Theory of Acceptance and Use of Technology (UTAUT) [19].

A global trend is the inclusion of sustainability in education, which is why there is an increasing emphasis on the acquisition, by both teachers and students, of a wide range of skills or attributes that contribute to academic success, particularly for the labor market. It is therefore believed that an institution with academic and labor success will be an institution with sustainability [20]. Higher education aims to develop in students the ability to see actions, problems, solutions, and consequences in a context that involves scientific, technical, and economic aspects, but it is necessary to integrate into these new concepts, for example, social responsibility and sustainable development in virtual environments [21].

A review of specialized literature on the subject of evaluation of LMS demonstrates the importance of previous research [2,8,10,22–25]. There are studies about online distributed programming that analyze students' perceptions [26]; however, no systematic quantitative studies that indicate a model that evaluates the acceptance of LMSs and that determine the factors that impact the performance and academic efficiency of students have been found. There are isolated studies on the subject of e-learning, LMSs, VLEs, and m-learning in different educational levels and training centers. Different models and theories have been used for evaluation, e.g., the TAM [27].

In a previous report, Varela et al. [28] noted that the use of the TAM has been of great support to predict when a system should be implemented, and they concluded that students, as they advance in their degree studies, increase their use of ICTs. It is necessary to highlight the constructs defined by the TAM:

1.  *Behavioral intention:* The plan to use or not use specific technology for learning;
2.  *Attitude towards using:* The negative or positive feeling of an individual about the accomplishment;
3.  *Perceived usefulness (PU):* The degree to which people believe that the use of the system will help them to achieve progress in the performance of their work;
4.  *Perceived ease of use (PEU):* The degree of ease associated with the use of the system or technology.

In another study, the construct [6,17,29] used was the VLE for which the TAM was applied, and the Anxiety (A), PU, and PEU variables were assessed, while the variables studied in the PLE were PU and PEU [2]. On the other hand, and taking into account that ICT has experienced a boom with the use of mobile devices in recent years, the m-learning model in LMSs has been studied using the TAM and the UTAUT and by taking A, Virtual Library (VL), PU, and System Factors (SF) as variables. In [22], it is concluded that the reality indicates that we do not have a full grasp of the aspects of acceptance of mobile use in education yet; hence, he describes that there is plenty of room for further research and contribution to the models for the application of technology in schools. On the other hand, studies indicate that the LMS must have support resources for students. One of these could be a VL, so students can consult the bibliographic collection for their courses. As one study points out, students using an e-learning method with a VL tend to frequently use it to study outside the university [30]. The study suggested that a well-designed online learning environment allows students to learn without having to use memorization by using an online collaboration model to solve simulated problems instead [31]. As is evidenced by the results of this study [32], to obtain strategies for online collaborative learning, it is necessary to use a combined method, such as a questionnaire, an interview, and content analysis, that is, Collaborative Learning Through the Computer (CLTC) [33].

A quantitative study was carried out in computer engineering programs in universities in Ecuador using evaluation factors, including Tools Learning (TL), to evaluate personal virtual environments, the percentage calculation of the use of the virtual classroom, and the learning resources: Google Docs, social networks, virtual meetings, and videos [2]. System Factors (SF), such as the use of the systems, the control of resources, technical knowledge, and the compatibility of using computer systems, were identified and evaluated [34]. This research indicates that the user must have a system administrator, as well as technical knowledge and computer equipment in virtual environments [18].

The access strategies (ASs) of the virtual systems or campus can be done using personal computers or mobile devices. Studies show that the ASs were personal computers and mobile devices, which were shared between students, as well as access addresses [32]. Using the TAM, TRA, and UTAUT models, the Mobile Use (MU) was evaluated for the implementation of m-learning, where it was highlighted that the use and utility in virtual education is prevailing with the use of cell phones [15].

A topic of high importance in the implementation of LMSs is the architecture to be used [35]. They indicate that the design, development, and maintenance of large virtual campuses that are used for e-learning are complicated topics. They describe the architecture of a sizeable multiplatform university campus. They analyzed three variables related to the architecture: (1) Software architecture, (2) detailed software design, and (3) the hardware architecture. The virtual campus works with various functions [36], which provide support for teaching through virtual forums, encourage teaching innovation, promote communication between different users, facilitate student tracking, self-learning, and self-evaluation, and provide teaching experiences mixed with varying degrees of virtuality.

When designing a virtual campus and virtual classrooms, these must include the software design, what the presentation and interaction with the modules will be like, and the hardware architecture with the specification of resources. Some studies propose a virtual campus design highlighting software design and hardware architecture [35]. The configuration must have information about the institution, and it must be well organized and easy for users to understand. The configuration of the classroom must attract users and must be intuitive and unambiguous; the interaction must occur through the connection of information within the LMS site, which allows access to content quickly and systematically [37].

In the same context of e-learning and LMS, studies performed at the Institute of Electrical and Electronics Engineers indicate that academic resources should use metadata, that is, Learning Objects with Metadata (LOM) [38]. While it is found the following metadata which can be used as: Name, author, title, and description. According to the rules of citation in IEEE, the date, format, and address of online storage (URL, Uniform Resource Locator) should be added to the previous ones. In the same line of research, studies have been presented to the scientific community using different LMSs:

- **Edmodo:** The use of Edmodo has allowed for evaluations of student learning through the elements that the corporate author of Edmodo facilitates on its platform, [39].
- **Schoology:** The objective of this free platform is to create a learning strategy for students and to motivate studying online. On this platform, teachers and students are able to create groups and courses, administer resources, set courses created as public or private, integrate resources from external platforms, and present statistics of the progress of each student [40].
- **Classroom:** As the author indicates, the speed of uploading files is perfect for working inside and outside the classroom. It is free for educational centers, designed for teachers and students, includes email, storage, and forums for collaboration between professors and students in the same class or different classes, and allows users to share or integrate videos, PDFs, and images. The user can also incorporate the URLs of virtual libraries or external resources. Iftakhar evaluated the use of Classroom by surveying students [41].
- **Moodle:** Moodle is an open-source LMS that allows users to tailor the design to each institution. In this study, the use percentage in undergraduate students was evaluated.

The scientific community has carried out evaluations in ICT implementation models, experimenting with student groups and making calculations using statistical coefficients. The support of e-learning with interactive virtual campuses, informative websites, and LMSs was evaluated individually with Cronbach's Alpha coefficients [42]. This was aimed at determining the level of reliability and the correlation coefficient and identifying correlation and the evaluated factors.

With statistical data, reliability is determined using the Cronbach coefficient, where the results show an acceptable trend when values above 70% are obtained. Another measure to validate research instruments is the reliability variable, the results of which, when close to 100%, are considered the most reliable [42]. Accuracy is the coefficient that allows evaluation of statistical correlation using estimates, where 7% is accurate, between 8% and 14% is acceptable, and between 15% and 20% is fair. This is the statistical method to evaluate an association between two continuous variables, which allows us to identify the relationship that exists between the variables and to determine which are direct or inverse, so as to determine the impact of the study variables [43]. Nevertheless, our proposal contains three coefficients and an identification of metadata. Different models and theories have been used for evaluation, for example, the TAM [19,29].

There have been isolated studies, such as the effect of Anxiety–Innovation (AI), Utility and Use (UU) [6], and learning tools [2], as well as SF, which relate emotions with learning tools qualitatively [34]. Some quantitative studies have been done using statistical tools, such as Cronbach reliability, Pearson variation, and Spearman correlation.

The main factors analyzed were AI, perceived use [6], online tool use [2], system control evaluation [34], accessibility of the system, library usage (Yang), and mobile usage (Ahmad). The objective of this study is to integrate all of these investigations and complement them with statistical coefficients to quantitatively determine academic efficiency in the use of the LMSs.

When pandemics such as that of COVID-19 occur, institutions should bear a share of social responsibility, and the use of the LMS Classroom is an excellent way for them to do so [44].

## 2. Methodology

The first stage consisted of determining the study LMS, which consisted of LC, LS, LE, and LM, since parts of their platforms are free. The second stage consisted of creating courses using subject sheets aimed for an average of 25 students per course. The third stage was the evaluation of the architecture, where the hardware and software were defined (Table 1). The fourth stage was the metadata, through which the learning objects were identified, as shown in Table 1, as well as academic resources. The fifth stage was the insertion of statistical coefficients, the Cronbach, the coefficient of variation, and correlation for the evaluation of metadata and factors. The sixth stage was the elaboration of questionnaires to quantify factors such as anxiety and innovation, utility and use,

and learning tools, as shown in Figure 1. Finally, in the eighth stage, through the statistical coefficients applied to the students through the questionnaires, the academic efficiency in relation to each LMS was determined.

*2.1. Configuration and Architecture*

Figure 1 shows the four LMSs used for this study: Edmodo, Google Classroom, Schoology, and Moodle.

The requirements for course configuration and architecture are found in Table 1.

1. **Institution and course registration:** University information and six courses (Figure 1) were registered:(1) OA, (2) SE, (3) SQD, (4) ITS, (5) ITPM, and(6) IC. All were applied to the educational program of engineering in information technology.

2. **Academic resources:** The virtual objects were created as suggested. They should be integrated into courses with LMSs; virtual objects should be created. In this study case, documents were created for each topic: (a) A PDF document with 10 sheets of content, (b) a Word document with 10 sheets of material, and (c) a 3–5 min video and a 10 slide presentation. Each LMS had a VL integrated with links to research work through a website that was shared in chat rooms and used Web 2.0 tools: Hangouts, Skype, and video chat. Each course had between one and two chat rooms. The virtual meetings were only with students for teachers to advise students.

3. **Virtual classes:** The activities were text analysis in discussion forums on the topic studied, a real case analysis where teams of four students were assigned, one final project, and three virtual meetings. The follow-up was carried out by the teacher, who evaluated the participation of students in chat rooms with forums for assistance and homework submission, giving feedback through messages in forums and advice in the virtual chatroom.

4. **Hardware:** The server for the Moodle LMS had a storage capacity of 480 GB and 8.0 GB RAM; in the cases of Edmodo, Google Classroom, and Schoology, the server used had 20 GB of storage. The transmission tower used for the four LMSs was provided by the region supplier. Four TP-Link Archer C9 type access points and 600 Mbps, 2.4 GHz, and 1300 Mbps, 5.0 GHz AC1900 wireless dual-band Gigabit Routers with three removable antennas were used. The workstations consisted of 75 computers with an i5 processor and with 8.0 RAM, a 500 GB hard drive, and an ethernet network card TP-link Tg 3468 at 10/100/100 Mbps and 32 bits.

5. **Software:** For the Moodle LMS, the Linux Centos version 2.7 operating system, a Mysql database with J2EE, and Windows FTP firewall software were used. The Internet used had a download speed of 4.0 Mb/s and an upload speed of 1.0 Mb/s. The browsers used were Internet Explorer 9.0, Mozilla 3.0, and Google Chrome for Edmodo, Google Classroom, and Schoology. The online software used is found at the following URLs: https://classroom.google.com, https://new.edmodo.com, and https://www.schoology.com.

6. **Types of users:** The user profile for the administrator is ideally a computer systems engineer with experience in the use of LMSs, a bachelor's degree, and experience in implementing courses through an LMS, and the ideal student for this test is one with a background in Information Technology Engineering.

*2.2. Statistical Metadata and Coefficients*

1. **Metadata characteristics:** The metadata for the Learning Object (LO) described by Wiley included name, author, description, initiative, and LOM, presented by date, format, and filing [45]. A metadata search was performed in each LMS. An LO search record was obtained for each course, and the categorization was done by the type of LOM.

2. **Statistical coefficients:** For the calculation of statistical coefficients, seven questionnaires were used, as shown in Table 2, with evaluation factors and reference authors.

**Table 1.** Configuration, architecture, and users in all four LMS.

| LMS Configuration | | | | Architecture Resources | | Users |
|---|---|---|---|---|---|---|
| *Registration of institution in LMS* | *Registry courses* | *Academic resources* | *Virtual classes* | *Hardware* | *Software* | *Types of users* |
| Registration of institution data:<br><br>Name, country, and sector Teacher/student data record: First name, country and College year | Course data: Name, semester, and period.<br>Style configuration Interface design Access code Inscription | Learning tools<br><br>Virtual libraries Virtual rooms | Assignment activities<br><br>Student follow-up Evaluations statistics | Server.<br><br>Transmission tower Access point Workstation | HTTPS, firewall, connection, internet, LMS platform, database, browser, mobile-app | Content manager in LMS.<br><br>Teacher User Students |

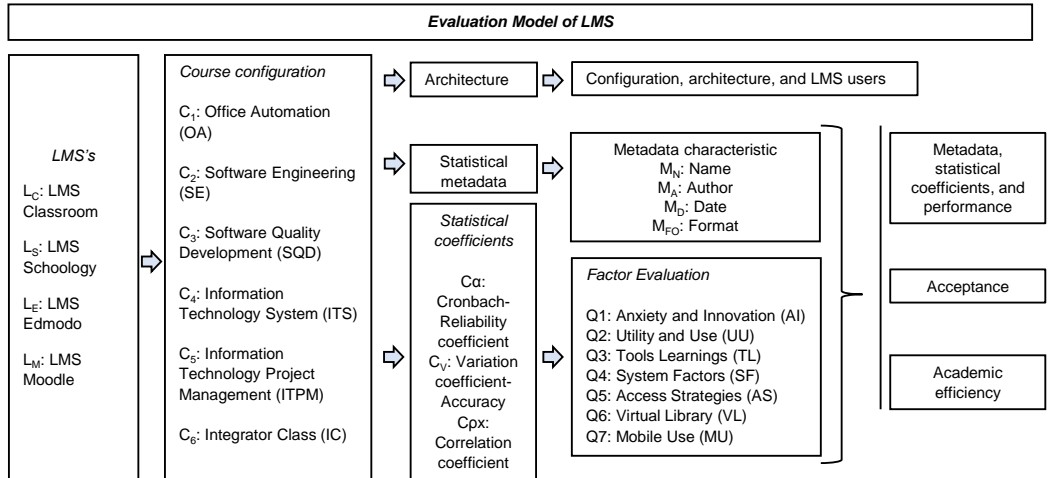

**Figure 1.** Learning Management System (LMS) factor models to estimate academic performance, academic acceptance, and academic efficiency.

**Table 2.** Questionnaires: Q1; (Anxiety–Intelligence (AI)), Q2; (Utility and Use (UU)), Q3; (Tools Learning (TL)), Q4; (System Factors (SF)), Q5; (Access Strategies (AS)), Q6; (Virtual Library (VL)), Q7; (Mobile Use (MU)).

| Questionnaire and Factor Evaluation | Items | Scale Answers | Coefficients Statistics |
|---|---|---|---|
| Q1-AI [6,19] | 10 | | |
| Q2-UU [29,34] | 11 | | |
| Q3-TL [2,34] | 11 | Likert type 1 to 5, where 1 is totally disagree, and 5 is strongly agree | Cronbach reliability. Pearson variation. Spearman correlation. |
| Q3-SF [19,46] | 8 | | |
| Q3-AS [32,47] | 23 | | |
| Q3-VL [10,30] | 9 | | |
| Q3-MU [22] | 4 | | |

Equation (1) was used to determine reliability.

$$\alpha = \frac{K}{K-1}\left[1 - \frac{\Sigma(s_i^2)}{s_T^2}\right],\tag{1}$$

where $\alpha$ is the Cronbach's alpha, and $K$ is the item number. $\Sigma(s_i^2)$ is the sum of variances of the items, and $s_T^2$ is the variance of the sum of items. $\alpha > 0.9$ is excellent, $\alpha > 0.8$ is good, $\alpha > 0.7$ is acceptable, $\alpha > 0.6$ is questionable, $\alpha > 0.5$ is poor, and $\alpha < 0.5$ is unacceptable.

Equation (2) was used to calculate accuracy.

$$C.V = \frac{\sigma}{\bar{x}},\tag{2}$$

where $\sigma$ is the standard deviation, and $\bar{x}$ is the arithmetic mean. A precision from 0 to 0.25 is accurate, 0.26 to 0.50 is acceptable, 0.51 to 0.75 is unacceptable, and 0.75 is inaccurate.

Equation (3) was used to calculate correlation.

$$\rho(x,y) = \frac{\sigma(x,y)}{\sigma(x)\sigma(y)},\tag{3}$$

where $\sigma(x,y)$ is the variance between $x$ and $y$, $\sigma(x)$ is the standard deviation of $x$, and $\sigma(y)$ is the standard deviation of $y$. Values close to 1 show a direct correlation, values very close to 0 indicate no correlation, and values close to $-1$ indicate an inverse correlation.

### 2.3. Factor Evaluation

Questionnaires were used for the seven evaluation factors that are presented in Table 3. The mathematical equation used is Equation (4).

$$\bar{x} = \frac{\Sigma x}{n},$$ (4)

where $\bar{x}$ is the arithmetic mean, $x$ is the numerical value of data group, and $\Sigma x$ is the sum of $x$.

**Table 3.** Averages of the metadata search by means of Learning Objects with Metadata (LOM), courses, and statistical coefficients in the LMSs.

| | LMS Metadata Search | | | | Metadata Types | | | | Statistical Coefficients | | | | | |
|---|---|---|---|---|---|---|---|---|---|---|---|---|---|---|
| | $L_S$ | $L_E$ | $L_C$ | $L_M$ | $M_N$ | $M_A$ | $M_D$ | $M_{Fo}$ | LMS | $C_\alpha$ | LMS | $C_V$ | LMS | $C_{\rho_x}$ |
| $C_1$ | **105** | 104 | **105** | 51 | **134** | 82 | 66 | 83 | $L_C$ | 0.73 | $L_E$ | 0.69 * | $L_E$ | $-0.46^*$ |
| $C_2$ | 57 | 83 | **98** | 53 | **165** | 39 | 45 | 42 | $L_S$ | **0.79 ** | $L_C$ | 0.32 | $L_E$ | $-0.22$ |
| $C_3$ | 45 | **62** | 60 | 36 | **124** | 26 | 5 | 48 | $L_C$ | 0.74 | $L_C$ | 0.29 | $L_S$ | 0.12 ** |
| $C_4$ | 45 | **62** | 60 | 36 | 63 | **76** | 6 | 12 | $L_S$ | 0.67 * | $L_M$ | 0.28 ** | $L_S$ | $-0.22$ |
| $C_5$ | 43 | 28 | **31** | 25 | **69** | 26 | 25 | 7 | $L_C$ | 0.72 | $L_M$ | 0.34 | $L_E$ | $-0.18$ |
| $C_6$ | 79 | 76 | **105** | 44 | **152** | 37 | 51 | 64 | $L_S$ | **0.75** | $L_E$ | **0.25 *** | $L_E$ | **0.17 *** |
| $\Sigma p$ | 374 * | 415 ** | **459 *** | 245 * | **707 *** | 286 ** | 198 * | 256 | | | | | | |
| % | 25% | 28% | **31%** | 16% | **49%** | 20% | 14% | 18% | | | | | | |

*** Highly significant value, ** very significant values, * significant values.

### 2.4. Academic Efficiency

The evaluation of the factors that had the greatest academic impact on students was carried out using the grades obtained in each course. Numerical grades that ranged from 0 to 10 for each student in the courses were used. The scale used for the grade point average was the following: 0 to 7.9, 8.0 to 8.5, 8.6 to 9.0, 9.1 to 9.5, and 9.6 to 10. The indicator was determined to obtain the three impact evaluation factors, following the logical condition described below:

1. $A > C$ and $A > B$ indicate a major impact $A$.
2. $B < A$ and $B > C$ indicate a normal impact $B$.
3. $C < B$ and $C < A$ indicate a minor impact $C$.

Considering the values of $A$, $B$, and $C$ in the courses $C_n$:
$C_1 \longrightarrow A(0.74 \text{ to } 0.89)$, $B(0.67 \text{ to } 0.85)$, $C(0.61 \text{ to } 0.82)$
$C_2 \longrightarrow A(0.70 \text{ to } 0.84)$, $B(0.63 \text{ to } 0.69)$, $C(0.61 \text{ to } 0.75)$
$C_3 \longrightarrow A(0.69 \text{ to } 0.83)$, $B(0.68 \text{ to } 0.74)$, $C(0.60 \text{ to } 0.76)$
$C_4 \longrightarrow A(0.80 \text{ to } 0.87)$, $B(0.77 \text{ to } 0.80)$, $C(0.72 \text{ to } 0.79)$
$C_5 \longrightarrow A( 0.73 \text{ to } 0.84)$, $B(0.67 \text{ to } 0.83)$, $C(0.60 \text{ to } 0.76)$
$C_6 \longrightarrow A(0.73 \text{ to } 0.84)$, $B(0.67 \text{ to } 0.73)$, $C(0.51 \text{ to } 0.70)$

## 3. Results and Discussion

### 3.1. Configuration, Architecture, and LMS Users

In our research, the configuration for each LMS was obtained when registering the institution and identifying the students' data in the four LMSs [35]. The first observation was that when the teacher shared the access code to the courses, 100% of the students were enrolled within the first three days. The organization of the virtual classes allowed students to manage their own schedules and activities, have a calendar in each course, and obtain feedback through virtual meetings.

### 3.2. Metadata, Statistical Coefficients, and Performance

The results in Table 3 exhibit the average metadata search in each LMS, the types of metadata, and the statistical coefficient.

The LMS metadata with the highest frequency were found in courses $C_1$ and $C_6$, with 105 for both $L_S$ and $L_C$ (Table 3). The highest number of searches (LMS metadata) was 459 in $L_C$, LOM being the most used by students. For $L_E$, the most considerable value obtained was 415. However, the students had a higher preference for $L_C$; for $L_S$, where a significance of 374 was obtained. Due to the difficulty with which the students looked for the LOM, search button visibility was not adequate. In the percentages representing access to information, the $L_C$ obtained was 31%. Likewise, for Ferran et al., access to academic resources was 54%, which is considered a very high value by the authors [48].

On the other hand, it was observed that, for the metadata types, the highest number of searches was for $M_N$ with a total of 707 queries, followed by $M_A$ with 286 and $M_D$ with 198. This $M_N$ result is because, for $C_2$ and $C_6$, the students preferred to search immediately by the name of the files. $M_A$ had a more significant presence; for $C_1$ and $C_4$, students had forum activities within the chat thread, and for them, searching by author facilitated the access. Amongst metadata types, $M_D$ had a lower nuumber of searches (198) than $M_N$, $M_A$, and $M_{Fo}$. This is because not all students were interested in looking for LO through $M_D$. Therefore, the total value was 198, and that accounted for 14% of the metadata types. In the study performed by Ferran et al., a result of 25% was obtained searching for LO by format, and it was determined that this was a high value [48].

In the case of $C_2$, $L_C$ obtained 98 searches, and $M_N$ obtained 165; this is because the students located the LOM through electronic presentations according to the topics of the course that facilitated the location in $L_C$. For the $C_3$, in $L_E$, 62 searches were obtained, with an $LOM$ of $M_N$ of 124, because in this course, the $M_N$ of the LOM was shown on the first screen of the $L_E$, and there are units of the $C_3$ with PDF file and electronic file presentation.

The $C_4$ obtained for $L_E$ was 62 in the LMS metadata search, and for metadata types, the most significant result was an $M_A$ of 76. This is because, in $C_4$, there were activities where the students located $M_A$ values that were authors, the teacher, and classmates so as to analyze information and obtain feedback. For $C_5$, in the search of metadata, the $L_S$ was 43, and the LOM used in $M_N$ was 69. This result was presented because the students participating in $C_5$ through $L_S$ had the mobile app, which gave them notifications of the files that had to be analyzed in the $C_5$ activities. For this group, 100% of the participants used a cell phone, as Ahmad recommended [15], which afforded students access to the $L_S$ so as to carry out the activities of the $C_5$. For the $C_6$, the search for metadata was an $L_C$ of 105, and the LOM used was an $M_N$ of 152. For this case, the students searched the video files and electronic presentations, and could access the $L_C$ with any device with Internet. Participants interacted among them, sharing the LOM through $M_N$.

Castro et al. obtained 60% and 50% of their elements used [49], and Solumou et al. concluded that the LOM is the most influential model to identify LOs in the learning process [38]. These values are comparable with this study, where we obtained 49% of LOM use with $M_N$, because students used their academic resources through the use of metadata. At the same time, Arciniegas et al. explained that LOM is the highest standard of metadata, and ongoing validity continues to validate that metadata are defined with the resource relationship of an educational process [47].

We also observed the results of the reliability, precision, and correlation coefficients of the four LMSs to determine the best performance among the six courses. The study using $C_\alpha$ (Cronbach model) has been widely discussed.

This study compared values with Humanante et al. [2]. $C_\alpha = 0.83$ for the constructs of the collaborative virtual environment and $C_\alpha = 0.84$ in communications, which showed excellent reliability, while $C_\alpha = 0.58$ in the level of consciousness, which was not reliable. For the investigation, the $C_\alpha$ was 0.79 for $L_C$ and 0.75 to $L_S$; for this study, this was considered as acceptable reliability according to the Cronbach scale. The $C_1$ course showed the highest $C_\alpha$ with the use of $L_S$; in this course, the students

manifested their acceptance to use $L_S$. For them, it was interesting to generate a learning environment through virtual media.

However, for the coefficient average of the four LMSs that presented reliability, the highest average is $L_C$ with a value of 0.73, which is to say, for the six courses, the LMS of Classroom was the one with the most significant impact. In $L_E$ and $L_M$, reliability was not presented, because not all courses received reliable acceptance according to the evaluation by the Cronbach coefficient. For $C_V$, $C_6$ was obtained with a precision of $L_E = 0.25$, and in $L_C$, an average of 0.30 was obtained. It obtained a lower average, which allows a precision for this LMS. This is because the $C_V$ variable indicates that the acceptable precision of a statistical study should not be higher than 30% for its accuracy. For the correlation coefficient $C_{\rho_x}$, the course that presented a satisfactory correlation is $C_6 = 0.17$. In this case, being positive, there is an acceptable direct correlation.

The averages obtained for each LMS with respect to the coefficients are the following: On average, an $L_C$ of $C_\alpha = 073$ is acceptable in this LMS. The students had the highest reliability by having their activities organized in their same space, in the cloud through the drive, and on their personalized site, where notifications of activities were in real time. $C_V = 0.305$ is accurate. The $C_{\rho_x}$ did not appear, which means that, for $L_C$, the relation of the questionnaires applied in the evaluation is independent. For $L_E$, the $C_\alpha$ was not acceptable: $C_V = 0.470$ and $C_{\rho_x} = -0.16$. In the case of $L_M$, the $C_\alpha$ did not appear: $C_V = 0.307$ and $C_{\rho_x} = -0.18$; for $L_S$ of $C_\alpha = 0.70$, $C_V$ was not presented: $C_{\rho_x} = -0.17$. The correlation presented in the LMS was very weak. Perez et al. indicated that, in terms of precision, they found that values higher than 0.25 were predominant, which indicates that it was due to the lack of error in the planning of the research [43]. In this investigation, they were also higher than 0.25, which indicates that a planning strategy for possible LMS errors should be put into place.

### 3.3. Acceptance

Our study presents the factors that influence the LMS in the higher-level institution that was most accepted by students of the six courses. Figure 2 shows the factors evaluated in the four LMSs.

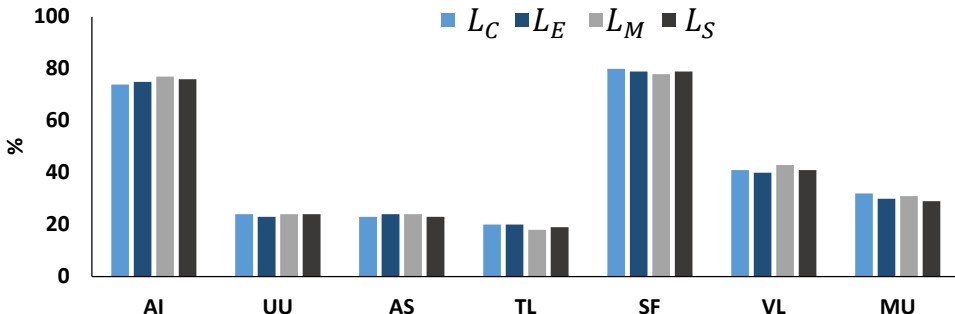

**Figure 2.** Results of evaluation factors for LMSs and the percentages of factors that influence the evaluation of LMSs for acceptance.

Figure 2 shows that $L_C$ had the most significant impact on *SF*, with a value of 82%, which was because of the positive reception by students when using system resources, and this result is consistent with the studies of Taylor et al. and Moore et al. [34,46]. They indicate that the compatibility and ease of access to the system have an impact on the users, with a value of 60%. The result of *AI* with 80% was because students had their first interaction with $L_C$. In contrast, Raaij and Schepers et al. obtained 82% vs. Davis et al. with 84% [6,19,29].

Innovation occurs when the user initiates an interaction for the first time, but the behavior may be stressful. Through this variable, Wiley determines that innovation occurs when a system is used. The next most influential factor was VL = 43%, which allowed students to have access to libraries with academic resources to help with their course study. Mee determined that both teachers and students

require that they are provided with a VL for online courses, obtaining 70% in the exclusive use of three libraries [30].

For $L_E$, SF accounted for about 80% because the app was installed on their cell phones or because the direct use of the Internet allowed easy access. AI accounted for 77% because it was the first interaction, and VL = 40% because it integrated the material into the courses. Similar values for $SF$ and $AI$ in $L_M$ and $L_S$, as well as for VL in LC and $L_E$, were found because the students evaluated the access to their courses more highly through an SF, but also showed a high percentage of AI, whereas VL was a third factor impacting their learning with the use of LMS. We compared our results with those of Ahn's, which indicated 50.5% of self-efficacy, as described in the self-confidence in the ability to archive results. The quality system accounted for 58%. For AI, this percentage was 77%, and for SF, it was 80% [50]. More recent studies (Park et al.) assess A at a 90% impact and associated it with a self-efficacy at 55% [51].

### 3.4. Academic Efficiency

Figure 3 illustrates the most significant impact factors on the courses for the evaluation of student academic efficiency.

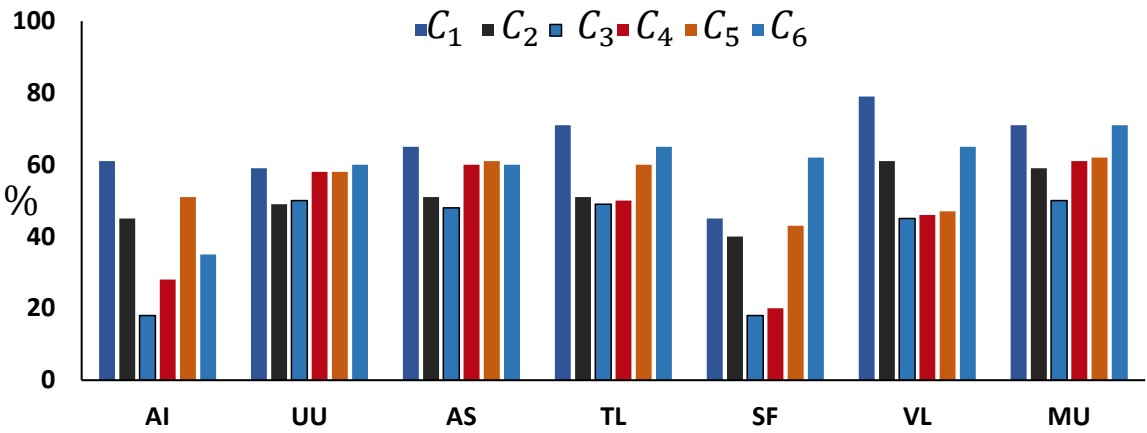

**Figure 3.** Results of evaluation factors in the six LMS Classroom courses.

As shown in Figure 3, six LMS Classroom courses were evaluated using seven factors for each of them, where the $C_1$ indicates that AI is above average, which is because the students were freshman students and had not worked before with any LMS. In the same course, the use of VL was the one with the most significant impact on students. It was essential for them to identify the VL to download school materials, books, articles from research journals, access scientific publication sites, and other resources necessary to back up their academic work. The three factors of higher percentage, TL, VL, and MU, indicate that, for the six courses, these should have had a TL, VL, and MU with an impact higher than 50%.

Previous studies indicate that the factors that show a minimum impact on academic efficiency are AI, SF, and UU. This is because, for AI, even though students interacted with Classroom for the first time, they had already used Moodle in other courses. SF had the lowest amount of AI, since it was believed that the LMS Classroom did not need to have system factors that were superior to those that the institution provided them. In this sense, they interacted and carried out their activities without any problems. In the case of the UU, they indicated that, due to their useful development, they did not perceive use with a high level of impact. Students considered that the UU is implicit in the MU for their access through their cell phone to the LMS.

In the case of $C_3$, the factors had a percentage lower than the average. The factors of a smaller percentage in this course were $AI$ and $SF$, and this is because, in the group, not everyone had access to

the Internet on their phones, and not all laboratory equipment had Internet access. However, in $C_6$, UU, AS, TL, SF, VL, and MU exceeded the average. Only the AI again had less impact on the students.

Therefore, we can affirm that the AI does not affect the performance of academic efficiency, since $VL$ has a more significant impact on students, and MU is a priority when using the LMSs in$C_L$ in the different courses.

These factors have the most significant impact concerning the evaluation obtained by the students in the six courses taught in the LMS.

Table 4 shows the results of the evaluation factors according to their impact with the evaluation scale used to measure student academic efficiency.

**Table 4.** Average of the evaluation results of factors in LMSs concerning the evaluation scales. Approved ≥ 8.0, not approved ≤ 7.9.

| Course | Scale Evaluation | AI | UU | AS | TL | SF | VL | MU |
|--------|-----------------|----|----|----|----|----|----|----|
| $C_1$ | | 0.76 | 0.55 | 0.63 | **0.89 \*\*\*** | 0.45 | **0.82 \*\*** | **0.85 \*** |
| $C_2$ | | 0.63 | **0.69** | 0.66 | 0.66 | 0.62 | **0.68** | **0.72** |
| $C_3$ | $10 - 9.6$ | 0.25 | **0.73** | **0.74** | 0.71 | 0.23 | 0.54 | **0.83** |
| $C_4$ | | 0.00 | 0.00 | 0.00 | 0.00 | 0.00 | 0.00 | 0.00 |
| $C_5$ | | 0.00 | 0.00 | 0.00 | 0.00 | 0.00 | 0.00 | 0.00 |
| $C_6$ | | 0.28 | **0.70** | 0.62 | **0.70** | 0.64 | **0.81** | **0.70** |
| $C_1$ | | 0.55 | 0.60 | **0.65** | **0.69** | 0.39 | **0.82 \*\*\*** | 0.58 |
| $C_2$ | | 0.55 | 0.59 | **0.67** | **0.63** | 0.41 | 0.54 | **0.77** |
| $C_3$ | $9.5 - 9.1$ | 0.47 | 0.62 | **0.72** | 0.37 | 0.22 | **0.71** | **0.80 \*** |
| $C_4$ | | 0.52 | 0.58 | 0.61 | 0.60 | 0.50 | **0.84 \*\*** | **0.73** |
| $C_5$ | | 0.00 | 0.00 | 0.00 | 0.00 | 0.00 | 0.00 | 0.00 |
| $C_6$ | | 0.41 | **0.69** | 0.63 | **0.75** | 0.61 | 0.64 | **0.77** |
| $C_1$ | | 0.59 | **0.64** | **0.67** | 0.63 | 0.50 | **0.76** | **0.76** |
| $C_2$ | | 0.53 | **0.59** | **0.61** | **0.61** | 0.44 | 0.51 | **0.73** |
| $C_3$ | $9.0 - 8.6$ | 0.31 | **0.68** | **0.69** | 0.58 | 0.24 | 0.60 | **0.69** |
| $C_4$ | | 0.58 | 0.57 | **0.60** | **0.60** | 0.40 | **0.67** | **0.73** |
| $C_5$ | | 0.24 | **0.78 \*** | 0.73 | 0.73 | 0.22 | **0.78** | **0.80 \*\*\*** |
| $C_6$ | | 0.38 | 0.63 | 0.62 | **0.74** | 0.61 | **0.66** | **0.79 \*\*** |
| $C_1$ | | 0.58 | 0.53 | 0.60 | **0.61** | 0.35 | **0.72** | **0.75** |
| $C_2$ | | 0.57 | 0.61 | **0.68** | **0.64** | 0.43 | 0.60 | **0.67** |
| $C_3$ | $8.5 - 8.0$ | 0.37 | **0.76** | 0.73 | **0.79 \*** | 0.32 | 0.65 | **0.80 \*\*** |
| $C_4$ | | 0.62 | 0.58 | 0.69 | **0.70** | 0.43 | **0.77** | **0.73** |
| $C_5$ | | 0.24 | **0.81 \*\*\*** | 0.71 | **0.72** | 0.30 | 0.70 | **0.77** |
| $C_6$ | | 0.46 | 0.56 | 0.65 | **0.65** | 0.60 | **0.75** | **0.80 \*\*** |
| $C_1$ | | 0.60 | 0.53 | 0.61 | **0.62** | 0.47 | **0.74** | **0.68** |
| $C_2$ | | 0.34 | 0.28 | **0.33** | **0.28** | 0.27 | 0.27 | **0.38** |
| $C_3$ | $7.9 - 0.0$ | 0.00 | 0.00 | 0.00 | 0.00 | 0.00 | 0.00 | 0.00 |
| $C_4$ | | 0.57 | 0.61 | **0.66** | 0.64 | 0.53 | **0.81 \*\*** | **0.68** |
| $C_5$ | | 0.36 | **0.80 \*** | 0.77 | **0.87 \*\*\*** | 0.23 | **0.80 \*** | 0.79 |
| $C_6$ | | 0.37 | 0.62 | 0.59 | **0.64** | 0.62 | **0.68** | **0.77** |

\*\*\* Highly significant value, \*\* very significant values, \* significant values.

For the scale obtained in the range between 10 and 9.6, TL, VL, and MU were greater than 80% in $C_1$. UU, VL, and MU had a similar result in $C_2$; likewise, UU, AS, and MU in $C_3$, $C_4$, and $C_5$ did not present results in this range, and the same applied to UU, TL, and MU in $C_6$. As the results point out, students prefer to have the TL to do their assignments and the VL to have access to academic material of the course through MU in person. In this study, it was verified that the students need to use their cell phones in order to do their assignments. For the range of 9.5–9, the most influential factor for $C_1$, $C_3$, and $C_4$ was VL; for the $C_5$ course, no evaluations in this range were obtained. This is because the students could not administer their deadlines, and their work was reviewed in a previous period. For factors ranging from 9.0 to 8.6, there was a more significant MU presence in the six courses

implemented. The next most significant factor was TL for all courses, and the UU had an acceptance value of 0.78 for the $C_5$, which is a significant value that indicates that students consider the utility and perception of the use of the LMS $L_C$ as a key impediment for a satisfactory academic achievement. For the 8.5–8.0 threshold, the most significant factors in $L_C$ were MU, TL, and VL, even though MU and TL presented very substantial and significant values in the six courses. *UU* in $C_5$ had a highly significant value, meaning students of the six courses considered that there must be at least MU, TL, and VL for them to obtain averages of the specified range.

When factors had a value ranging from 7.9 to 0.0, the course was considered to be within a non-approved range. The factors TL, VL, and MU imply that students used these items slightly more than the rest, but it was observed in the course $C_3$ that no students in this range were affected in their academic efficiency, meaning all students passed the course.

Previous studies using the TAM indicate that the factors that the TAM uses should continue to be investigated, since academic efficiency is not evaluated, but the users' perceptions of the acceptance of the use of this technology are still being evaluated [52,53]. Herein, more factors were analyzed that allow for the integration of the evaluation of academic efficiency with the use of software for learning or training, and a model was developed in which there are measurement factors for academic efficiency. While Yu et al. studied the effects of learning by interviewing with students, the result was that forums in which they could send questions and receive feedback from teachers and classmates were of great help for their learning experience [54].

As we have previously mentioned, LMSs offer a great affordable alternative to face-to-face classes through free virtual classes when school activities have to cease in the case of a state of emergency, such as the one prompted by the COVID-19 pandemic. The results showed good reception of such methods, with 31% of the students preferring LMSs such as the LC (Table 3) over traditional classes. A pandemic such as COVID-19 involves taking physical distancing measures between teachers and students, which does not necessarily imply that school activities have to stop. As shown in this research, all six of the courses proposed were able to be conducted virtually. Virtual education using TLs, such as Zoom groups, Facebook, etc., is evidently all the more efficient when we take a look at results such as those shown in Table 4, where 89% of the most efficient students used these tools, and within this group, 85% used MU and 82% used VLs.

## 4. Conclusions

In summary, this study demonstrates that, for an LMS to yield the best results, SF, both hardware and software, must be considered for the architecture design, meaning that the server must have a capacity greater than 20 GB, with a minimum internet speed of 4 MB. The above-described metadata should also be taken into account for the search of information in LMS resources to be higher than 40%. Regarding the performance of the evaluation indicators in the period from September 2017 to August 2018, it is evident that Classroom had the highest reliability with an average of more than 60% for the evaluation period, but Schoology had an average of 79% during the period from September to December 2017 for the SE course, which was due to the involvement of social networks and chat sessions among participants. The precision indicator showed a decreasing trend for Edmodo ranging from 0.69% in the Office automation course. Precision for the SE course was 0.25%, since the use of chat rooms showed the effectiveness of Edmodo, which had a correlation value of 0.17, while for Schoology this was 0.12. This indicator showed the most stable behavior when measured for the use of the LMS, as is clearly evidenced by the interaction between the participants and the use of learning tools outside the course platform.

Regarding the level of acceptance, it was evident that the SF variable is mainly used to increase the use of the LMS and that their application in the four LMSs was higher than 80%; on the other hand, the AI measured in users did not exceed 80%, which indicates that the emotional state of the participants did not affect their performance, but innovative tools such as VLs did. When several virtual libraries were available for the LMSs, they allowed access to material different from the one in

the course platform for students studying their subjects. During our study, more than 50% of the 30 virtual libraries available were visited. Learning Tools had a more significant impact, 85% in academic efficiency, indicating that tools such as social networks, chat rooms, and video tutorials are of great help for students. VLs and MU yielded similar results, both being greater than 80%. We also found that VLs were visited by more than 90% of the students, and mobile use was over 90%, which led to increased academic efficiency, with students obtaining an average score of 9.0 when evaluated, evidencing that the virtual libraries and the use of mobile devices is key when studying using LMSs.

We can therefore conclude that academic efficiency in the evaluated system relied heavily on TL, MU, and VL, with 89%, 85%, and 82%, respectively. We believe that it is of the utmost importance that institutions have TL, such as virtual rooms, social media workgroups, storage repositories, video editing applications, and online documents. For MU, on the other hand, cell phones, tablets, and iPads would be of use. Finally, VLs such as EBsco, Google Academic, Latindex, and the IEEE browser are ideal resources. Regarding the human factor, the most outstanding students (average between 9.6 and 10) had the highest AI at 76% in the course $C_1$, probably due to the fact that this was the first interaction that students had with the LMS systems. In all six courses, the activities carried out by the outstanding students in courses $C_4$ and $C_5$ did not generate AI in them (0%). We also found that the use of cell phones made the teaching–learning process more dynamic, e.g., students would immediately read their notifications to check their assignments and instantly plan their activities, which was further evidenced by the results of the academic impact of MU, which was over 50% in all six courses. This model can be implemented internationally, and perceived use and utility for teachers using LMS can also be measured. Higher education institutions could largely benefit from the implementation of LMSs. From a social standpoint, universities would be able to better train individuals, develop knowledge, and share updated information that can be accessed at any time. Carrying out the activities through virtual courses eliminates the use of paper completely, and this, in turn, reduces deforestation, considering that large technology companies, such as Facebook, Google, and Microsoft, among others, are making modifications to achieve the sustainability of their servers through renewable energy sources, namely wind, solar, and blue energy, along with the reduction of environmental pollution in terms of transportation, electricity, construction, and maintenance. Implementing LMSs could set a reference for sustainability.

Finally, one of the takeaways from this research is that LMSs are as sustainable as they are affordable, since teachers and students only need a Gmail account to use e-learning and m-learning applications on their mobile phones, which are mostly available for free. This could lead to more institutions with a minimum of classrooms and, for users, the reduction of time and money allocated to transport.

The present investigation was carried out in a public university in Mexico. Currently, private institutions such as Tec de Monterrey and Universidad del Valle de Mexico use some of these e-learning tools. We believe it would be useful to perform further studies in these private institutions to see if they yield similar results for the factors under study. A new area of opportunity is the application of this methodology to teachers (the human factor) who use virtual tools to quantify learning development, perceived use, and utility.

**Author Contributions:** Conceptualization, B.J.S. and J.M.O.R.; methodology, J.R.-R., J.M.O.R., and J.E.E.G.-D.; software, J.M.O.R. and A.D.; validation, J.E.E.G.-D., J.R.-R., R.G.G., J.M.O.R., and B.J.S.; formal analysis, B.J.S., J.E.E.G.-D., J.R.R., A.D., and F.F.S.; writing—original draft preparation, J.E.E.G.-D., A.D., J.M.O.R., and J.R.-R.; writing—review and editing, B.J.S.; supervision, F.F.S., J.R.-R., and J.M.O.R. All authors have read and agreed to the published version of the manuscript.

**Funding:** This research was funded by the Consejo Nacional de Ciencia y Tecnología (CONACYT) and PRODEP.

**Acknowledgments:** The authors of this work are especially grateful to the Faculty of Engineering of the Autonomous University of Querétaro for their support in carrying out this research.

**Conflicts of Interest:** The authors declare that there is no conflict of interest.

## Abbreviations

| Variable | Description | Abbreviator | Description |
|---|---|---|---|
| $C_{\rho_x}$ | Correlation coefficient | A | Anxiety |
| $C_1$ | Office Automation (OA) | AI | Anxiety–Innovation |
| $C_2$ | Software Engineering (SE) | app | Application Program for a Particular |
| $C_3$ | Software Quality Development (SQD) | AS | Access Strategies |
| $C_4$ | Information Technology System (ITS) | $C_1$ | office automation |
| $C_5$ | Information Technology Project Management (ITPM) | CLTC | Collaborative Learning Through the Computer |
| $C_6$ | Integrator Class (IC). | IC | Integrator Class |
| $C_V$ | Variation coefficient—Accuracy | ICT | Information and Communication Technologies |
| $C_\alpha$ | Cronbach Reliability coefficient | ITPM | Information Technology Project Management |
| $K$ | Item number | ITS | Information Technology System |
| $L_C$ | LMS Classroom | LMS | Learning Management System |
| $L_E$ | LMS Edmodo | LO | Learning Object |
| $L_M$ | LMS Moodle | LOM | Learning Objects with Metadata |
| $L_S$ | LMS Schoology | MU | Mobile Use |
| $M_A$ | Metadata Author | PEU | Perceived Ease of Use |
| $M_D$ | Metadata Date | PLE | Personal Learning Environments |
| $M_{F0}$ | Metadata Format | PU | Perceived Usefulness |
| $M_N$ | Metadata Name | SE | Software Engineering |
| $n$ | Total number of $x$ | SF | System Factors |
| Q1 | Anxiety and Innovation (AI) | SQD | Software Quality Development |
| Q2 | Utility and Use (UU) | TAM | Technology Acceptance Model |
| Q3 | Tools learnings (TL) | TL | Tools Learnings |
| Q4 | System Factors (SF) | TRA | Theory of Reasoned Action |
| Q5 | Access Strategies (AS) | UTAUT | Unified Theory of Acceptance and Use of Technology |
| Q6 | Virtual Library (VL) | UU | Utility and Use |
| Q7 | Mobile Use (MU) | VL | Virtual Library |
| $S_i^2$ | Sum of variances of the items | VLE | Virtual Learning Environment |
| $S_T^2$ | Variance of the sum of items | | |
| $x$ | Numerical value of data group | | |
| $\bar{x}$ | Arithmetic mean | | |
| $\alpha$ | Reliability Alpha Cronbach | | |
| $\sigma$ | Standard deviation | | |
| $\sigma(x)$ | Standard deviation of $x$ | | |
| $\sigma(x,y)$ | Covariance between $xy$ | | |
| $\sigma(y)$ | Standard deviation of $y$ | | |

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
