# Peer review of "Learning Management System-Based Evaluation to Determine Academic Efficiency Performance"

_sustainability, doi:10.3390/su12104256_

Round 1

Reviewer 1 Report

The paper is interesting and current, in terms of subject matter, as it is the great advance that is generating more and more, virtual education. However, it presents a series of considerable limitations linked, firstly, to an eminently software-based study and the measurement of software use through the different applications by users in LMS. However, no reference is made to the context in which the study is carried out (brief description of the degrees, origin of this virtual teaching system, the roles of the teaching staff and students, in which university this has been worked on, country; the origin of this new teaching model, its trajectory, the procedure for dealing with it, etc.).

The title does not correspond with the content, for example, the term "sustainability" is not sufficiently justified or developed in all the discourse, only at the end when it refers to saving costs with this type of model, making use of the mobile and avoiding displacements, typical of the presence.

Secondly, although the bibliography is varied, it requires greater depth at the educational level (uses, applications, teacher training, etc.) and is more up-to-date. Only a reference from 2020 and the interval from 2017 to 2019 are not provided with bibliographical evidence.

Finally, the conclusions are too short and seem to be an extension of the discussion, providing percentage data; they are absent of specific implications, not only for the field of data engineering, but also for human factors (teachers and students), methodologies, type of activities, resources that promote greater dynamism, self -learning and social interaction, motivation, learning, proposals for improvement and foresight, regarding future lines of researching.

Author Response

Dear Reviewer

We are pleased to resend you the entitled article Sustainability in Learning Management System-based Evaluation to Determine the Academic Efficiency Performance for your consideration, with the good intention to be published in the section Special Issue "Sustainable Educational Management for Effective E-Learning"

Authors appreciate the corrections and hope that the answers enlisted below in this document fulfill the expectations. We considered every suggestion as a valuable opportunity to improve and enrich enormously our work. Therefore, we have intended to be as punctual as we could to attend the decision of Major Revisions. Reviewer observations are highlighted using bold black font, our replies using blue font and changes in existing sentences have been written in italic font.

  1. The paper is interesting and current, in terms of subject matter, as it is the great advance that is generating more and more virtual education. However, it presents a series of considerable limitations linked, firstly, to an eminently software-based study and the measurement of software use through the different applications by users in LMS.

However, no reference is made to the context in which the study is carried out (brief description of the degrees, origin of this virtual teaching system, the roles of the teaching staff and students, in which university this has been worked on, country; the origin of this new teaching model, its trajectory, the procedure for dealing with it, etc.)

We appreciate your comments to enrich the scientific content of our work. In the new manuscript a brief context of the study has been depicted as follows in order to make a brief description of certain factors of the research.

Line 32-34

“At the University of Chile, a study measured the level of educational and technological use of Moodle and its implications for teaching from a quantitative approach and applying a questionnaire to a sample of 640 teachers of Higher Education [12].”

Line 35-37

“In six universities in Saudi Arabia, studies were carried out to determine the importance of people's  attitudes  when they use LMS, in the teaching-learning process, showing that attitude is the main barrier to overcome in the implementation of LMS [13].”

Line 37-38

“In India there are increases, its universities adopt the LMS system, to take advantage of 700 virtual tools as seen in [14].”

Line 39-41

“Humanante evaluated at the National University of Chimborazo - Ecuador, the computer engineering program [2]. Learning tools for personal virtual environments, factors influencing the environment were evaluated in Jordan by Ahmad [5].”

Line 41-45

“At the University of China, the confirmation and perception of the students and the data through 45 Chinese participants, were analyzed in an executive postgraduate program, resulting in the conceptual construction model to explain the individual differences between students in the level of acceptance and use of its Virtual Learning Environment (VLE) [6].”

Line 45-47

“In Iraqi universities, LMS systems have been adopted to improve the educational processes in higher education institutions, through Employment Technology Acceptance Model (TAM) To Adopt Learning Management System (LMS) [16].”

  1. The title does not correspond with the content, for example, the term "sustainability" is not sufficiently justified or developed in all the discourse, only at the end when it refers to saving costs with this type of model, making use of the mobile and avoiding displacements, typical of the presence.

Thank you for the feedback. The latest version of the paper, the title has been changed to:

Learning Management System-based Evaluation to Determine the Academic Efficiency Performance

 in order to match the content of the manuscript.

Besides, the next lines regarding sustainability were added.

Line 48-54

“A global trend is the insertion of sustainability in education, which is why there is an increasing emphasis on the acquisition, by teachers and students, of a wide range of skills or attributes that contribute to academic success, particularly in the labor market, that is, an institution with academic and labor success will be an institution with sustainability [17]. High school education is the ability to see actions, problems, solutions and consequences that include scientific, technical and economic aspects, however it is necessary to involve new concepts such as social responsibility and sustainable development in virtual environments [18].”

Also, the next paragraph was inserted to explain the contribution of this work to sustainability.

Line 464-473

“In higher education institutions in the social part, the contribution is in the development of knowledge, which allows universities to share information and train society with updated information that can be accessed at any time, through courses implemented in LMS, and carrying out the activities through virtual courses, eliminates the use of paper completely, and the result impacts the reduction of deforestation. It is important to consider essential criteria that are essential in favor; large technology companies, such as Facebook, Google, Microsoft, among others, are making modifications to achieve sustainability of their servers through renewable energy sources, taking advantage of the benefits of wind, solar and blue energy. Environment pollution in terms of transportation, electricity costs, building maintenance costs, and introduction integrates a reference to sustainability.”

  1. Secondly, although the bibliography is varied, it requires greater depth at the educational level (uses, applications, teacher training, etc.) and is more up-to-date. Only a reference from 2020 and the interval from 2017 to 2019 are not provided with bibliographical evidence.

Thank you for your comments. The following references have been added to have more contemporary state-of-the-art Kiennert et al. [11]; Daradoumis et al. [23]; Polizzi et al. et al. [5]; Luet al.  [7]; Magalhãeset al. [1]; Denan et al. [4]; Cabero-Almenara et al. [12]; Alghamdi et al. [13]; Radif et al. [16]; Gulzar et al. [14].

  1. Finally, the conclusions are too short and seem to be an extension of the discussion, providing percentage data; they are absent of specific implications, not only for the field of data engineering, but also for human factors (teachers and students), methodologies, type of activities, resources that promote greater dynamism, self -learning and social interaction, motivation, learning, proposals for improvement and foresight, regarding future lines of researching.

Thank you for the observation. We are include in the latest version:

Line 453-464

“The specific implications, the academic efficiency in the evaluated system depend mainly on the TL, MU, and VL, with 89%, 85%, and 82% corresponding. So, a specific implication is an importance that the institution has TL; tales like virtual rooms, social media workgroups, storage repositories, video editing applications, online documents, For MU, the most common are cell phones, tablets, Ipad. Finally, the VLs can be E-Bosco, Google Academic, Latindex, and the IEEE browser. Regarding the human factor, the most outstanding students (average between 9.6 and 10) the highest AI with 76% was presented in course C1, because it is the first interaction that students had with LMS systems. During the six courses, the activities carried out by the outstanding students not generated AI (0%) in the teaching of courses C4 and C5. The use of cell phones for the teaching-learning process caused dynamism, because immediately, to read their notifications and plan their activities, that is why it is presented, values ​​of 50% in the six courses. This model can be implemented internationally, and perceived use and utility for teachers using LMS can also be measured.”

Reviewer 2 Report

This is a report of evaluation of a model, using four Learning Management System (LMS), through the design of architecture, the configuration, metadata, and statistical coefficients.

There are several issues to be amended before consideration of this paper as suitable for publication in the journal.

In the abstract section:

Authors report: “integrating the factors that other 6 researchers have used only models isolation,” : it seems like comparison here: authors should rephrase. Also, there is an introduction/background covering half of the abstract. This section, although unstructured should equally contain all sections of an abstract.

Authors report: “has the highest level of performance, with an average of 73% when evaluated using statistical coefficients”: compared with what? And the 73% percentage is an efficient one in such models? There are models presenting a higher accuracy level when operating.

Authors should explain the rationale of referring to COVID, as their aim was not that, in “Institutions must comply with a social responsibility, when pandemics like COVID-19 occur, the use of the LMS Classroom is an excellent proposal.”.

In the Introduction section:

Authors should also explain: what are the “no systematic quantitative studies” in comparison with isolated studies?

There are a lot of technical abbreviations such as ICT, LMS, VLE that the reader is hard to follow. I would suggest authors to use only some of them (such as LMS) and follow the term throughout the whole text.

There is an extended paragraph referring to studies, but without aim. Authors should briefly in this section describe the existing literature, the gap and the rationale of their study.

In their methodology, authors should be more narrative and describe in a more clear way their methodology, not only implementing technical terms, for example using terms from the first paragraph of the results section.

In the results section, authors provide with comparisons with other studies: these can be moved to the discussion section.

There must be a synchronization between all sections, including the discussion, which has to be reformatted in a more structured way to reflect the exact findings of the study. Limitations and direct comparisons should be added, in terms of efficacy and accuracy levels of their system.

Author Response

Dear Reviewer

We are pleased to resend you the entitled article Sustainability in Learning Management System-based Evaluation to Determine the Academic Efficiency Performance for your consideration, with the good intention to be published in the section Special Issue "Sustainable Educational Management for Effective E-Learning"

Authors appreciate the corrections and hope that the answers enlisted below in this document fulfill the expectations. We considered every suggestion as a valuable opportunity to improve and enrich enormously our work. Therefore, we have intended to be as punctual as we could to attend the decision of Major Revisions. Reviewer observations are highlighted using bold black font, our replies using blue font and changes in existing sentences have been written in italic font.

Reviewer

This is a report of evaluation of a model, using four Learning Management System (LMS), through the design of architecture, the configuration, metadata, and statistical coefficients.

There are several issues to be amended before consideration of this paper as suitable for publication in the journal.

1.- In the abstract section:

Authors report: “integrating the factors that other 6 researchers have used only models isolation,” : it seems like comparison here: authors should rephrase. Also, there is an introduction/background covering half of the abstract. This section, although unstructured should equally contain all sections of an abstract.

Thank you for your observations. The subjects in the abstract are included in the introduction section as:

Line 112-115

“In the design of a virtual campus and virtual classrooms, they must include; the software design, how will be the presentation and interaction with the modules, the hardware architecture, with the specification of resources. Some studies propose a virtual campus design highlighting software design and hardware architecture [35].”

Line 115-118

“The configuration must have information about the institution, and it must be organized and easy for users to understand. The configuration of the classroom must attract users, must be understandable and unambiguous; the interaction must occur through the connection of information within the LMS site, which allows access to content quickly and systematically [37].”

 Line 91-97

“The evaluation factors include: TL to evaluate personal virtual environments, in Ecuadorian universities, for the computer engineering program, a quantitative investigation was carried out; the percentage calculation of the use of the virtual classroom, and the learning resources that considered were; using google documents, social networks, virtual meetings and videos [2]. The SF: were identified and evaluated through the factors that intervene in the use of the systems, in the control of resources, technical knowledge and the compatibility of using computer systems [34].  Research indicates that you must have a system administrator, as well as technical knowledge and computer equipment   [26].”

Line 98-100

“The AS of the virtual systems or campus can be done using personal computers or mobile devices. Studies show that the access strategies were; personal computers and mobile devices, which were shared between students as well as access addresses [32]”.

Line 100-102

“Using the TAM, TRA, and UTAUT model, he evaluated the MU, for the implementation of m. learning, where he highlighted that the use and utility in virtual education is prevailing with the use of cell phones [15].”

Line 146 -155

“The statistical data, reliability is determined using the Cronbach coefficient, where the results show an acceptable trend when values above 70% are obtained. Another measure to validate the instruments of an investigation is the reliability variable, where the results close to 100% are the most reliable [42]. The accuracy: The coefficient that allows evaluating statistical quality using estimates, where 7% is accurate, between 8% and 14% it is acceptable, and between 15% and 20% is fair, correlation, is the statistical method to evaluate an association between two continuous variables, which allows us to identify the relationship that exists between the variables, and to determine which are of direct or inverse correlation, to know the influence of the study variables [43.]

2.- Authors report: “has the highest level of performance, with an average of 73% when evaluated using statistical coefficients”: compared with what?

Thank you for your feedback. Table 3 depicts the highest level of the performance. if you average the values of 73%, 74% and 72% corresponding to LC, the result is 73%, which is the Cronbach coefficient, for which when you have results higher than 70%, the result is already reliable according to the Cronbach theory.

The statistical results of Cronbach are reliable for LC and LS because both presented greater reliability of 70% (73% for LC and 73.6% for LS). After all, the virtual tools of both are similar.

3.- And the 73% percentage is an efficient one in such models?

Thank you for your feedback. 73% is an efficient one according to the Cronbach model according to [42].

Please could you see:

  • “MATRIX FOR ESTIMATING ADEQUACY OF INTERNAL CONSISTENCY COEFFICIENTS WITH RESEARCH MEASURES”

Ref: [Ponterotto, J. G., & Ruckdeschel, D. E. (2007). An overview of coefficient alpha and a reliability matrix for estimating adequacy of internal consistency coefficients with psychological research measures. Perceptual and motor skills, 105(3), 997-1014.]

  • George and Mallery (2003) provide the following rules of thumb: “_ > .9 – Excellent, _ > .8 – Good, _ > .7 – Acceptable, _ > .6 – Questionable, _ > .5 – Poor, and _ < .5 – Unacceptable”

Ref: [Gliem, J. A., & Gliem, R. R. (2003). Calculating, interpreting, and reporting Cronbach’s alpha reliability coefficient for Likert-type scales. Midwest Research-to-Practice Conference in Adult, Continuing, and Community Education.]

 4.- There are models presenting a higher accuracy level when operating.

We appreciate the comment. In the introduction section, we updated the phrase as:

Line 153-55

“Nevertheless, our proposal contains 3 coefficients and identification of metadata. Different models and theories have used for the evaluation, for example, the use of the Technology Acceptance Model (TAM) [27,29]”

5.- Authors should explain the rationale of referring to COVID, as their aim was not that, in “Institutions must comply with a social responsibility, when pandemics like COVID-19 occur, the use of the LMS Classroom is an excellent proposal.”.

Thank you so much. We wanted to describe contemporary issues. However, we  rewrote the sentence as

Line 163-164

“Institutions must comply with a social responsibility, when pandemics like COVID-19 occur, the

 use of the LMS Classroom is an excellent proposal [44].”

[44] Zhang, W.; Wang, Y.; Yang, L.; Wang, C. Suspending Classes Without Stopping Learning: China’s Education Emergency Management Policy in the COVID-19 Outbreak. J. Risk Financial Manag. 2020, 13, 55.

6.- In the Introduction section:

Authors should also explain: what are the “no systematic quantitative studies” in comparison with isolated studies?

We appreciate the comment. In the latest version of the paper we are introducing qualitative and quantitative studies. Thus, the next items are threated along the paper:

in the present investigation:

Quantitative studies:

  • Evaluation of models LMS
  • Statistical metadata
  • statistical coefficients
  • Academic efficient

Qualitative studies:

  • Configuration of course
  • Architecture
  • configuration, architecture and LMS users
  • Metadata characteristic

Line 156-158

“There have been isolated studies such as the effect of AI, UU [6], and learning tools [2], as well as SF, which allow relating emotions and learning tools qualitatively [34]. Some quantitative studies were used through statistical tools such as: Cronbach reliability, Pearson variation, Spearman correlation.”

7.- There are a lot of technical abbreviations such as ICT, LMS, VLE that the reader is hard to follow. I would suggest authors use only some of them (such as LMS) and follow the term throughout the whole text.

We appreciate your feedback. We have an abbreviation section at the end of the manuscript. However, in the introduction section we wrote:

An abbreviation section is included to make it more suitable for reading using the e-learning argot.

We also believe that the variable titles will be repeated a lot, that is why the abbreviations were made, developing a glossary that was integrated at the end of the investigation.

8.- There is an extended paragraph referring to studies, but without aim. Authors should briefly in this section describe the existing literature, the gap and the rationale of their study.

We appreciate your comments. In the latest version of the manuscript we are pointing out the following existing literature to relationate the study:

Line 159-162

“The main factors analyzed are: AI, perceived use, and use [6], online tools [2]. Systems control evaluation [34]. Accessibility to the system. Meet library usage [Yang], and mobile usage [Ahmad]. The objective of this study is to integrate all these investigations and complement them with statistical coefficients to determine academic efficiency in the use of the LMS quantitatively.”

9.- In their methodology, authors should be more narrative and describe in a more clear way their methodology, not only implementing technical terms, for example using terms from the first paragraph of the results section.

Thank you for your kind comments. Currently, we are including more narrative to describe clearly technical aspects. However, in the introduction section we wrote: “An abbreviation section is included to make it more suitable for reading using the e-learning argot”. to make it easier to follow.

Also, in subsection 2.1 we are implying

Line 166-175

“The first stage consisted of determining the study LMS, which were LC, LS, LE, LM, since part of its platform, is free. The second stage was the construction of the courses for which they used, subject sheets, and an average of 25 students per course. The third stage was the evaluation of the architecture, where the hardware and software were defined (Table 1). The fourth stage is the metadata through which the learning objects identified, as shown in Table 1, Academic resources. The fifth stage was the insertion of statistical coefficients; Cronbach, coefficient of variation, and correlation, for the evaluation of metadata and factors. The sixth stage was the elaboration of questionnaires to determine the factors: A, Usefulness, and use, etc., as shown in Figure 1. Finally, in the eighth stage, through the statistical coefficients applied to the students through the questionnaires, the academic efficiency was determined related to each LMS.”

10.- In the results section, authors provide with comparisons with other studies: these can be moved to the discussion section.

We appreciate your kind comments. We placed the comparison in the Results and Discussion sections as you suggested.

Line 256

“3. Results and Discussion”

The results were linked to the discussion. So, the reader has fresh ideas of the results and discussion.

11.- There must be a synchronization between all sections, including the discussion, which has to be reformatted in a more structured way to reflect the exact findings of the study.

We appreciate your observation. We renamed the section from Results to “Results and Discussion” to have a formatted structure to reflect the exact findings of our study.

12.- Limitations and direct comparisons should be added, in terms of efficacy and accuracy levels of their system.

Thank you for your kind feedback. We added direct comparison as (Lines 479-483)

“The present investigation was carried out in a public university in Mexico. Currently, private institutions such as Tec de Monterrey and Universidad del Valle de Mexico use some tools. It would be convenient to disseminate the results in these private institutions to see if they present similar factors, TL, VL, and MU. A new area of opportunity is the application of this methodology to teachers (human factor) who use virtual tools for learning development, perceived use, and utility.”

Round 2

Reviewer 1 Report

The changes suggested have been added in the final paper, improving considerably the study.